# Clinical Origin and Species Distribution of *Fusarium* spp. Isolates Identified by Molecular Sequencing and Mass Spectrometry: A European Multicenter Hospital Prospective Study

**DOI:** 10.3390/jof7040246

**Published:** 2021-03-25

**Authors:** Anne-Cécile Normand, Sébastien Imbert, Sophie Brun, Abdullah M. S. Al-Hatmi, Erja Chryssanthou, Sophie Cassaing, Christine Schuttler, Lilia Hasseine, Caroline Mahinc, Damien Costa, Christine Bonnal, Stéphane Ranque, Marc Sautour, Elisa Rubio, Laurence Delhaes, Arnaud Riat, Boualem Sendid, Lise Kristensen, Marcel Brandenberger, Juliette Guitard, Ann Packeu, Renaud Piarroux, Arnaud Fekkar

**Affiliations:** 1AP-HP, Groupe Hospitalier La Pitié-Salpêtrière, Service de Parasitologie Mycologie, 75013 Paris, France; sebastien.imbert@chu-bordeaux.fr (S.I.); renaud.piarroux@aphp.fr (R.P.); arnaud.fekkar@aphp.fr (A.F.); 2AP-HP, Hôpital Avicenne, Service de Parasitologie-Mycologie, 93000 Bobigny, France; sophie.brun@aphp.fr; 3Centre of Expertise in Mycology Radboud University Medical Centre, Canisius Wilhelmina Hospital, 6525 Nijmegen, The Netherlands; a.alhatmi@unizwa.edu.om; 4Natural & Medical Sciences Research Center, Department of Microbiology, University of Nizwa, Nizwa 616, Oman; 5Division of Clinical Microbiology, Karolinska Institutet, Department of Laboratory Medicine, 171 77 Stockholm, Sweden; erja.chryssanthou@sll.se; 6CHU Toulouse, Service de Parasitologie-Mycologie, 31000 Toulouse, France; cassaing.s@chu-toulouse.fr; 7Laboratoire Biogroup-LCD, 92300 Levallois, France; C.schuttler@biogroup-lcd.fr; 8CHU de Nice, Service de Parasitologie Mycologie, 06200 Nice, France; HASSEINE.L@chu-nice.fr; 9CHU de Saint Etienne, Service de Parasitologie Mycologie, 42000 Saint Etienne, France; Caroline.Mahinc@chu-st-etienne.fr; 10Centre Hospitalier Universitaire de Rouen, Service de Parasitologie Mycologie, 76000 Rouen, France; damien.costa@chu-rouen.fr; 11AP-HP, Hôpital Bichat-Claude Bernard, Service de Parasitologie Mycologie, 75018 Paris, France; christine.bonnal@aphp.fr; 12Aix Marseille University, IRD, AP-HM, SSA, VITROME, IHU Méditerranée Infection, 13005 Marseille, France; stephane.ranque@ap-hm.fr; 13CHU de Dijon, Service de Parasitologie Mycologie, 21079 Dijon, France; marc.sautour@u-bourgogne.fr; 14Department of Microbiology, ISGlobal Barcelona Institute for Global Health, Barcelona, Spain CDB, Hospital Clinic, University of Barcelona, 08036 Barcelona, Spain; ELRUBIO@clinic.cat; 15CHU de Bordeaux, Groupe Hospitalier Pellegrin, Service de Mycologie, 33404 Bordeaux, France; laurence.delhaes@chu-bordeaux.fr; 16Laboratory of bacteriology, Division of Laboratory Medicine, University Hospital of Geneva, Rue Gabrielle-Perret-Gentil 4, 1205 Genève, Switzerland; Arnaud.Riat@hcuge.ch; 17Department Parasitology-Mycology, CHU de Lille, 59000 Lille, France; bsendid@univ-lille2.fr; 18Department of Clinical Microbiology, Aarhus University Hospital, 8200 Aarhus, Denmark; lise.kristensen@rm.dk; 19Labor Synlab Luzern, 6002 Lucerne, Switzerland; Marcel.Brandenberger@synlab.com; 20Centre de Recherche Saint-Antoine, Inserm, CRSA, AP-HP, Hôpital Saint-Antoine, Service de Parasitologie-Mycologie, Sorbonne Université, 75012 Paris, France; juliette.guitard@aphp.fr; 21Sciensano, BCCM/IHEM collection, Mycology and Aerobiology Unit, 1000 Brussels, Belgium; Ann.Packeu@sciensano.be; 22Inserm, Institut Pierre Louis d’Epidemiologie et de Santé Publique, Sorbonne Université, 75571 Paris, France; 23Inserm, CNRS, Centre d’Immunologie et des Maladies Infectieuses, Cimi-Paris, Sorbonne Université, 75005 Paris, France

**Keywords:** fusariosis, mALDI ToF mass spectrometry, DNA sequencing, elongation factor, *Fusarium* species complex

## Abstract

*Fusarium *spp. are widespread environmental fungi as well as pathogens that can affect plants, animals and humans. Yet the epidemiology of human fusariosis is still cloudy due to the rapidly evolving taxonomy. The Mass Spectrometry Identification database (MSI) has been developed since 2017 in order to allow a fast, accurate and free-access identification of fungi by matrix-assisted laser desorption ionization—time of flight (MALDI-TOF) mass spectrometry. Taking advantage of the MSI database user network, we aim to study the species distribution of *Fusarium *spp. isolates in an international multicenter prospective study. This study also allowed the assessment of the abilities of miscellaneous techniques to identify *Fusarium* isolates at the species level. The identification was performed by PCR-sequencing and phylogenic-tree approach. Both methods are used as gold standard for the evaluation of mass spectrometry. Identification at the species complex was satisfactory for all the tested methods. However, identification at the species level was more challenging and only 32% of the isolates were correctly identified with the National Center for Biotechnology Information (NCBI) DNA database, 20% with the Bruker MS database and 43% with the two MSI databases. Improvement of the mass spectrometry database is still needed to enable precise identification at the species level of any *Fusarium* isolates encountered either in human pathology or in the environment.

## 1. Introduction

Fungal species belonging to the *Fusarium* genus have the tremendous ability to colonize extremely different environments from soil to miscellaneous living organisms, and as far as the dining table of the international space station [1]. They are therefore responsible for infections in plants, animals and humans and are also responsible for the degradation of organic matter including books, fabrics or paintings [2]. In humans, *Fusarium *spp. illnesses encompass a variety of superficial, invasive or disseminated pathologies called fusariosis. Fusariosis affect both immunocompetent and immunocompromised people, the latter being at risk of developing invasive diseases. Recently, a pulmonary superinfection due to *Fusarium proliferatum* has also been described in a patient with severe COVID-19 [3]. To date, the genus *Fusarium* comprises a large number of species, which are themselves distributed within species complexes (SC) sharing common traits. The difficulties in properly understanding the epidemiology of *Fusarium* lie in the constant evolution of the taxonomic classification and in the difficulties of easily and accurately identifying a given species. Fast and accurate identification at the species level or at least at the species complex level is also important from a medical point of view as it is a prerequisite for optimal management of infections to warrant the appropriate antifungal drug initiation. As an example, most human invasive fusariosis is caused by *Fusarium solani* species complex (FSSC) and *Fusarium oxysporum* species complex (FOSC) but the first complex is more resistant to amphotericin B and voriconazole than the second [4].

In the past few years, MALDI-TOF (matrix-assisted laser desorption ionization—time of flight) mass spectrometry has become a leading tool for the identification of microorganisms including pathogenic fungi. While a DNA sequence-based identification can be considered as the reference, it is not within the reach of all microbiology laboratories. On the contrary, MALDI-TOF MS is now well developed and a lot of laboratories are now equipped with a device allowing an easy, fast and accurate identification of microorganisms, including fungi. Nevertheless, a robust and exhaustive database that encloses less common fungal genera and species is a prerequisite for a correct identification. In this perspective, in 2017 we developed an online open free database for identification of fungal agents, the Mass Spectrometer Identification database (MSI) [5], that is now used by approximately 200 laboratories all over the world.

Due to the relative scarcity of *Fusarium* infections, relevant studies reporting fusariosis epidemiology data are rare, retrospective, and most required data collection over a large period to obtain enough isolates and relevant conclusions [6,7]. Therefore, we decided to perform a prospective analysis to describe the species’ clinical distribution of *Fusarium *spp. from different centers that use the Mass Spectrometry Identification (MSI) platform and to compare the mass spectrometry and molecular approach for identification of *Fusarium* isolates.

## 2. Materials and Methods

Collection of *Fusarium *spp. isolates. The study was conducted during a one year period from 1st January to 31st December 2018. Isolates were selected from the MSI database user statistics extraction. MSI users are known to have various needs of the application. To gain a better understanding of the epidemiology of fusariosis, centers that use the application for all their fungal identifications were asked to participate in the study. Hence, once a month, a list of isolates corresponding to *Fusarium *spp. was established and the participating centers were asked to provide information (such as the type of sample, whether it was a superficial or invasive fusariosis, and the overall diagnosis regarding the patient disease) and to send their isolates to the La Pitié-Salpêtrière Mycology laboratory for further analyses.

Categorization of clinical implication. According to the patient clinical outcome and the sample origins, four categories were designed: colonization (broncho-aspirations, bronchoalveolar lavage, cutaneous and mucosal samples, sputum and stools); invasive fusariosis (biopsies, blood samples and stools with a documented colitis); onychomycosis (nails); and keratitis (contact lens, contact lens liquid, cornea, conjunctiva and eye vitreous).

Mass spectrometry identification. After subculture on Malt agar, mass spectrometry identification was checked using the Bruker database (RUO—Research Use Only—and filamentous fungi combined) that contains 15 *Fusarium/Acremonium* references, the online MSI application that contains 50 *Fusarium/Acremonium* references and the updated version of the MSI application that was released in 2020 (https://msi.happy-dev.fr/, accessed on 22 January 2020) and includes 57 *Fusarium/Acremonium* references. Extraction protocol was performed as previously described [8].

Molecular identification. Molecular identification was performed by sequencing part of the translation elongation factor 1α (TEF1 α) using the following primers: EF1 (ATG GGT AAG GAR GAC AAG AC) and EF2 (GGA RGT ACC AGT SAT CAT GTT) [9]. Identification was then obtained by a DNA phylogenetic tree-based approach, using a previous one built by the Dutch Fungal Biodiversity Centre (Westerdijk Fungal Biodiversity Institute, The Netherlands) [10] according to the maximum likelihood method. We then chose this method as the gold standard for identification. Sequences were first aligned with the MAFFT program. The alignment was then imported into Mega v6.2. Then 95 sequences from reference strains owed by the CBS were added to our sequences for the construction of the phylogenetic tree. The sequence of the *Fusarium dimerum* LC177279.1 isolate was used as outgroup. Finally, a BLAST (Basic Local Alignment Search Tool) against the NCBI (National Center for Biotechnology Information) database was also performed. The best results with an identity score above 99% were considered as the identification given by the NCBI application. If several species obtained the same percentage of identity score, identifications were separated by the max score column.

## 3. Results

### 3.1. Weight of Fusarium spp. Among Filamentous Fungi

The study involved twenty centers located in Europe: 13 in France, two in Sweden, two in Switzerland, one in Denmark, one in Belgium and one in Spain (Figure 1 and Appendix A).

The study involves MSI’s users who acknowledge submitting all of their mold isolates to the MSI application. Hence, the number of spectra that they submit to the application represents a large proportion of the total submitted spectra by all users. During the one-year study period 66,590 spectra (which has a different meaning than the number of isolates, see below) were submitted to the MSI application and coupled with an identification score above the defined threshold of 20. Among these spectra, 44,733 (67.2%) corresponded to filamentous fungi (including dermatophytes) of which 1960 spectra (4.4% of the filamentous fungi) were *Fusarium *spp. The 20 participating centers accounted for 67.7% of the filamentous identifications produced by MSI (30,275 spectra) over this period and for 77% (1511 spectra) of the *Fusarium* identifications.

### 3.2. Origin of the Fusarium spp. Isolates

Each MSI user identified a single fungal isolate by making between one and four deposits on the MALDI-TOF plate. By having access to this information (how much deposit for an isolate) for each center, we were able to calculate the number of isolates based on the number of spectra submitted to the MSI database. Thus the 1511 spectra generated by the partners corresponded to the number of 471 *Fusarium* isolates for identification of which 205 were sent to the La Pitié-Salpêtrière Parasitology-Mycology laboratory.

Finally, 182 isolates were available for complete analyses as some isolates failed to grow again after subculture.

Clinical origin was various. Among the 160 isolates for which the information was available, most of the isolates (*n* = 56, 35%) were reported to cause superficial skin infection or respiratory colonization while 50 (31.25%) were responsible for onychomycosis, 37 (23.13%) caused keratitis and 17 (10.62%) were reported to cause invasive fusariosis.

Some differences were observed in terms of clinical origin per species complex, e.g., the *Fusarium fujikuroi* species complex represented nine of 17 invasive fusariosis cases but only 2 of 50 reported onychomycosis (*p* < 0.0001 by Fisher’s exact test).

The distribution of clinical forms by complex and species is presented in Table 1.

### 3.3. Identification of Fusarium spp. Isolates by Phylogenetic Tree Building Approach

The 182 DNA sequences of TEF1 α were first compared one to another. This allows for the definition of 86 different molecular patterns. These 86 patterns were divided as follows: 58 corresponded to a unique sequence only found once and 28 were clusters that included between 2 and 30 isolates each (for a total of 124 sequences) (Appendix A).

These 86 different sequences were submitted to the phylogenetic-based identification process. This led to the identification of 27 different Fusarium species belonging to seven species complexes (SC), namely *Fusarium oxysporum* (FO-SC), *Fusarium fujikuroi* (FF-SC), *Fusarium solani* (FS-SC), *Fusarium dimerum* (FD-SC), *Fusarium incarnatum/equiseti* (FIE-SC), *Fusarium sambuccinum* (FSa-SC) and *Fusarium redolens* (FR-SC) (Figure 2).

The overall repartition of these seven complexes for all participating centers is presented in Figure 3.

Three species complexes are largely represented and account altogether for nearly 95% of the total number of isolates. These are the *Fusarium oxysporum*-SC (*n* = 65, 35.7%), *Fusarium fujikuroi*-SC (*n* = 54, 29.7%) and *Fusarium solani*-SC (*n* = 53, 29.1%).

When focusing at the species level, *Fusarium veterinarium*, a recently described species from the *Fusarium oxysporum*-SC [11] was the most preponderant species (*n* = 45, 24.7%) and was found in 17 out of the 20 centers. The two other main found species were *Fusarium proliferatum* (*n* = 39, 21.4%, 12 centers) from the *Fusarium fujikuroi*-SC and *Fusarium petroliphilum* (*n* = 14, 7.7%, 7 centers) from the *Fusarium solani*-SC.

Interestingly, among the *Fusarium solani*-SC, the phylogenetic tree analysis revealed two new species: one of them (*F. solani*.new.sp.2) corresponding to eight isolates collected in seven centers, the second (*F. solani*.new.sp.1) represented by two isolates in two centers (Figure 2).

Thus, the molecular identification based on phylogenic tree was considered as the gold standard for *Fusarium *spp. identification in this study and used for the further assessment of NCBI-BLAST identification and mass spectrometry databases.

### 3.4. Identification by NCBI-BLAST Approach

Using the BLAST approach, none of the 65 isolates from the *Fusarium oxysporum*-SC were correctly identified at the species level using the NCBI database. All were identified as *F. oxysporum sensu stricto*, consequently correctly at the “species complex” level.

Regarding the *Fusarium fujikuroi*-SC, 94.4% (*n* = 51/54) of the isolates were correctly identified at the species level. Three misidentifications were observed, which concerned three *Fusarium sacchari* isolates that matched against a reference of *F. oxysporum.var.cubense* (KM263189.1). This is undoubtedly a mistake in the reference as, when blasted against NCBI, this latter matched exclusively with references from the *Fusarium fujikuroi*-SC, most of them being *F. sacchari* references.

Regarding the *Fusarium solani*-SC, 20.8% (*n* = 11/53) of the isolates were correctly identified at the species level as *Fusarium solani sensu stricto*. Isolates from other species (including *F. petroliphilum*, *F. keratoplasticum*, *F. lichenicola*, *F. falciforme*, and *F. solani*.new.sp.2) were identified as *F. solani* while isolates belonging to the new species *F. solani*.new.sp.1 were identified as *F. falciforme* by NCBI BLAST (Table 2).

Finally, the eight isolates belonging to *Fusarium dimerum*-SC, *Fusarium incarnatum*-SC, *Fusarium redolens*-SC and *Fusarium sambuccinum*-SC were correctly identified at the species level.

### 3.5. Identification by Mass Spectrometry

The accuracy of MALDI-TOF mass spectrometry was then assessed for the identification of the 182 *Fusarium *spp. isolates. The results of the identification performances according to three different mass spectrometry databases are presented in Table 2. Concordant results at the complex level (correct SC) and at the species level (correct SP), as well as discordant identifications and under threshold identifications are shown for the 27 species identified by the phylogenetic tree building approach.

As the seven species identified with the phylogenetic tree approach and belonging to the *Fusarium oxysporum*-SC were not yet described when the various mass spectrometry databases were built, no correct identification at the species level was obtained for the isolates of this complex. Hence, Bruker MALDI bio-typer (MBT), MSI-1 and MSI-2 databases allowed a correct identification of the *Fusarium oxysporum*-SC isolates at the species complex level for 66.1%, 78.5% and 98.5%, respectively.

Regarding the *Fusarium fujikuroi*-SC, MSI databases allowed a correct identification for 89% of the isolates at the species level and 100% at the complex level while a correct identification was possible for only 60% of the isolates at the complex level and 54% at the species level using the Bruker MBT database.

Regarding the *Fusarium solani*-SC, the identification rate at the species level varied from one species to another. The only correct identifications at the species level with Bruker MBT were obtained for three out of the 14 *Fusarium petroliphilum* isolates while this database allowed the identification of 57% of the isolates at the complex level. All of the 53 isolates were identified at the complex level with the two MSI databases, however, the identification at the species level depended on the species. Hence, *F. petroliphilum* (86% with MSI-1, and 100% with MSI-2), *F. lichenicola* (100%), *F. keratoplasticum* (75%) and *F. falciforme* (38%) obtained scores concordant with identifications at the species level with the two MSI databases.

Whatever the MS database used for identification, isolates belonging to *Fusarium dimerum*-SC were correctly identified. *Fusarium redolens*-SC and *Fusarium sambuccinum*-SC were not represented in either of the databases; hence, the species could not be identified by the MS approach.

## 4. Discussion

Our study shed light on the species distribution of *Fusarium *spp. isolates originating from clinical samples and confirmed the usefulness of mass spectrometry for accurate identification of uncommon fungal pathogens. Although the diversity among this genus is very important, more than 95% of the isolates were represented by only three species. Strikingly, there were some differences in distribution according to clinical origin.

Our study has several limitations. First, a possible bias of selection can be noticed, as only isolates that were first identified via MSI application with a score above the defined threshold of 20 were included. Hence, *Fusarium* isolates for which the identification threshold was not reached might have been overlooked. However, we gave the opportunity to the participating centers to send us any isolates they wanted to, whether or not they were identified with the MSI application, as long as they were analyzed during the study period. Each MSI user is free to use the application as he wishes. It should be noted that there was no duplicate among the 182 isolates that were analyzed. Secondly, there may be a bias in the exact number of isolates and the percentage of *Fusarium* among all molds. Indeed, we extrapolated the overall numbers of isolates from the number of spectra submitted. However, in some cases, a user may have identified the same isolate several times, at various ages of culture. Finally, although this was a prospective study, a substantial number of isolates was unavailable due to failures in shipment of strains or unsuccessful subculture.

The accurate species-level identification of *Fusarium *spp. isolates is still a matter of concern while fungal taxonomy is in perpetual evolution. This makes it even more difficult to obtain a precise, correct and reliable identification and can make even recently published studies obsolete. Moreover, it potentially generates disagreements between specialists and complicates the task of microbiologists who are not experts in the field, as well as clinicians in charge of patients. Finally, it makes it difficult to collect reliable epidemiological data. For example, some species that belong to the *Fusarium solani*-SC have been moved in 2019 into the genus *Neocosmospora* [12] while several mycologists recently issued a comment on this particular publication to deny this separation into two genera [13]. This example illustrates the difficulty for both microbiologists and clinicians in keeping up with the evolution of the fungal taxonomy. This point is also very well illustrated by considering a study similar to ours, published in 2009, and focusing on the identification of *Fusarium *spp. by mass spectrometry. This study included nine species whereas a decade later, our work includes 24 different species within seven species complex [14]. In the same vein, none of the 65 isolates identified by mass spectrometry in the present work as *Fusarium oxysporum* can further be considered as such since most of them (45/65) now correspond to *Fusarium veterinarium*, a newly described species [11]. Looking back over a few decades and considering that in 1940 Snyder advised mycologists to consider as *F. oxysporum* all species of the former section *Elegans*, and to attribute them forms depending on the host [15], we can see how far we have come in the complexification of the taxonomy of *Fusarium*. As *F. oxysporum* is a well-known human pathogen among clinicians, the epi-typification of the complex of species by Lombard et al. into 21 species (six of which not being named yet) and eight clades will make clinical interpretation difficult for fungal infections, especially if it is established that *F. oxysporum sensu stricto* is not implicated in human infections.

*Fusarium veterinarium* was named as such because it was mostly isolated from veterinary samples (e.g., shark peritoneum or canine stomach). In our case, we found *F. veterinarium* isolates to be implicated in human fusariosis, mostly keratitis (15/38), and onychomycosis (13/38). In a recent study by Najafzadeh et al. regarding *Fusarium* diversity in Iran [16], *F. veterinarium* was only identified as an agent of keratitis.

The second issue encountered during this study is the fact that we obtained ten isolates that could be considered as two not yet described species. These isolates are distributed in the *Fusarium solani*-species complex. These two new potential species were represented by several isolates recovered from various hospitals and were implicated in various clinical contexts: three were responsible for keratitis, three for onychomycosis, and 3 were considered as clinically non relevant cutaneous carriage. One case was not clinically documented.

Our set of 182 *Fusarium *spp. isolates was submitted to several identification processes and compared to a TEFα sequences-based phylogenetic tree building approach as gold standard. TEFα is one of the most reliable genes, along with the second largest subunit of RNA polymerase II gene (RPB2), for *Fusarium* species polymorphism study, and is deemed sufficient by specialists for phylogenetic studies of the genus [9]. The submission of the 182 DNA TEFα sequences to the NCBI database gave correct identification at the species complex level for 98.4% of the isolates (179/182). For three *F. sacchari* isolates, NCBI proposed *F. oxysporum* as a first identification, with the same identification percentage as *F. sacchari*, but a larger coverage. These misidentifications highlighted a labeling error in NCBI database as *F. oxysporum.var.cubense* (KM263189.1). This particular reference matches against our three isolates and is highly similar to many sequences of *F. sacchari* in the NCBI reference database. Identifications at the species level varied from 0%–100% depending on the species complex taken into account, *Fusarium oxysporum*-SC and *Fusarium solani*-SC being the less likely to be correctly identified with the NCBI-BLAST approach.

Generally speaking, the identification of microorganisms by mass spectrometry depends on the completeness of the database used. In a previous study, correct identification was obtained for 91.9% of isolates (57 among 62) with an in-house database [14]. In our work, the ability of the mass spectrometry approach to give an accurate identification showed important variations according to the database used. Correct identification at the species complex level could be achieved for only 61% of the isolates with the Bruker MBT database whereas it reached 91% using MSI-1 database and up to 98% using the upgraded MSI-2 database. Identification rates at the species level were lower than 50% whatever the MS database, except for the *Fusarium fujikuroi*-SC that had already benefited from an improvement on the MSI databases thanks to a study in collaboration with the Westerdijk Fungal Biodiversity Institute in 2015 [17]. Obviously, the global performances and the rate of correct identifications are directly related to the presence or absence of a given species in the database, and to the number of references corresponding [18]. Consequently, the improvement of the current MSI-2 database with references obtained from our 182 isolates will allow an improvement in identification at the species level.

Finally, our study gives an insight into the weight of *Fusarium *spp. within the world of human fungal pathogens. Within this genus, it gives a view of the clinical distribution of species and allows us to understand the limits of molecular biology and mass spectrometry for obtaining a precise identification, which is an essential prerequisite for carrying out relevant projects. In this sense, this work should encourage the development of collaborative studies focusing in particular on the clinical entities of fusariosis and the susceptibility of *Fusarium *spp. to antifungal drugs.

## 5. Conclusions

Within a complex and rapidly evolving taxonomy, the use of mass spectrometry and particularly the MSI online application is a fast and reliable tool for the accurate identification of *Fusarium *spp. isolates. However, improvement of the mass spectrometry database is also a dynamic process and significant progress still needs to be made to enable precise identification of any species encountered either in human pathology or in the environment. This must go hand in hand with a taxonomic clarification of the genus *Fusarium*, a broader vision of the epidemiology of fusariosis and a better knowledge of the impact of the species on the therapeutic management of patients.

## Figures and Tables

**Figure 1 jof-07-00246-f001:**
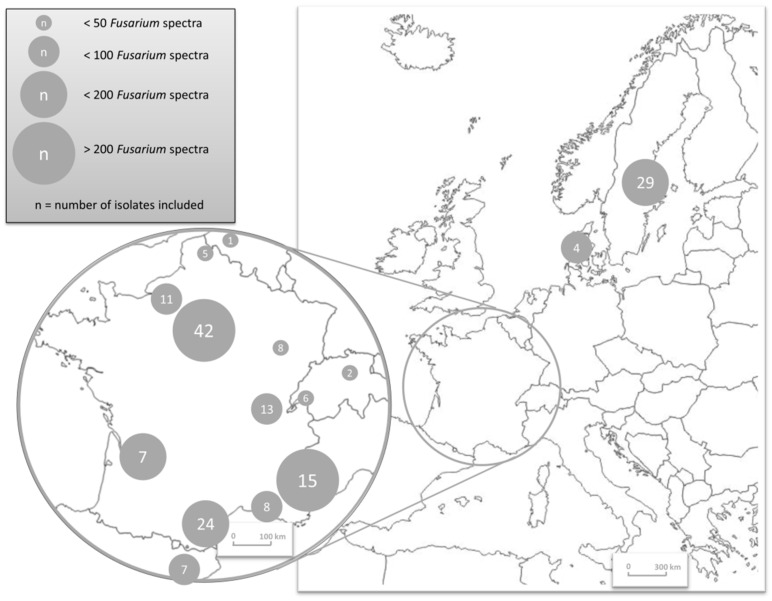
Geographical repartition of the 20 participating centers, and respective contribution to the project as indicated by the number of spectra sent to the Mass Spectrometry Identification (MSI) database.

**Figure 2 jof-07-00246-f002:**
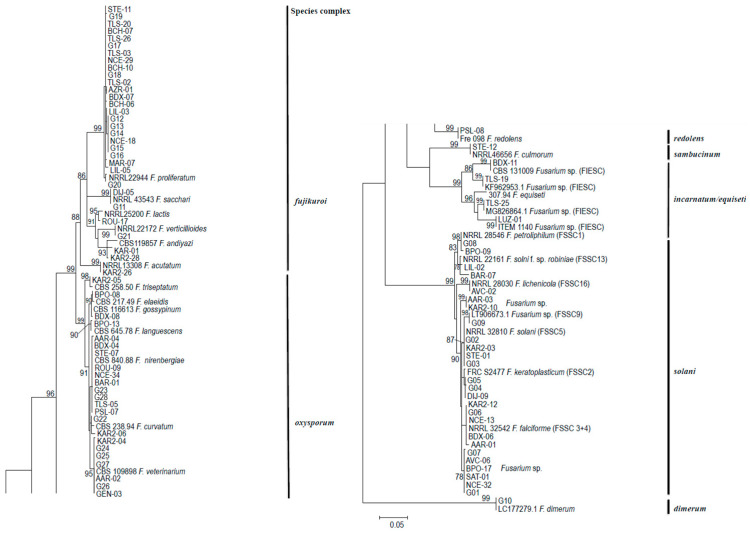
Distribution of 86 different sequences of TEF1 α from 182 Fusarium spp. isolates collected in 20 European centers during a one-year prospective study compared to reference sequences.

**Figure 3 jof-07-00246-f003:**
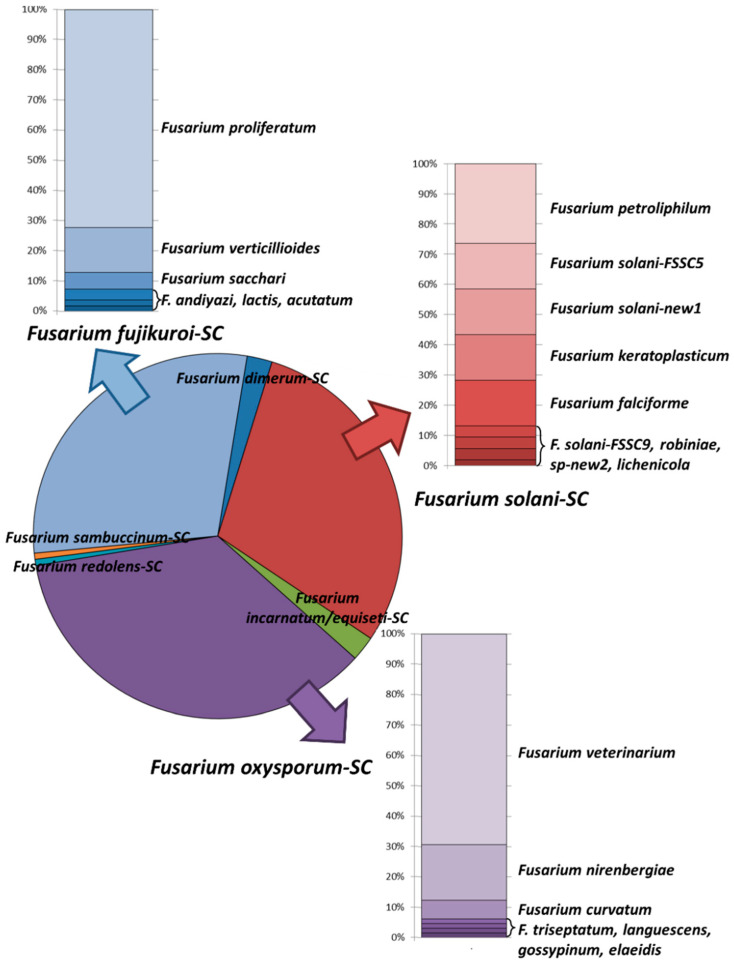
Species distribution of 182 *Fusarium *spp. isolates identified in 20 European centers centers during a one-year prospective study.

**Table 1 jof-07-00246-t001:** Distribution of 160 clinical *Fusarium *spp. isolates collected in 20 European centers during a one-year prospective study within the different species complexes and species according to the clinical form.

Species Complexes (Gold-Standard Identifications) Related Disease (Number of Cases)	Invasive Fusariosis	Keratitis	Onychomycosis	Colonization
*Fusarium oxysporum*-Species complex (sample origin documented *n* = 54/65)				
Total FO-SC	1	17	26	10
*Fusarium curvatum*			3	1
*Fusarium elaeidis*			1	
*Fusarium gossypinum*			1	
*Fusarium languescens*			1	
*Fusarium nirenbergiae*		2	7	
*Fusarium veterinarium*	1	15	13	9
*Fusarium fujikuroi-*Species complex (sample origin documented *n* = 48/54)				
Total FF-SC	6	9	2	31
*Fusarium acutatum*		1		
*Fusarium andiyazi*				1
*Fusarium proliferatum*	2	7	2	25
*Fusarium sacchari*	2			1
*Fusarium verticillioides*	2	1		4
*Fusarium solani-*Species complex (sample origin documented *n* = 50/53)				
Total FS-SC	1	9	20	20
*Fusarium falciforme*		1	3	3
*Fusarium keratoplasticum*		1	2	5
*Fusarium lichenicola*				1
*Fusarium petroliphilum*	1	3	8	2
*Fusarium solani new-sp1*		1	3	3
*Fusarium solani new-sp2*		2		
*Fusarium solani.sp.robiniae*				1
*Fusarium solani-FSSC5*		1	2	5
*Fusarium solani-FSSC9*			2	
Other-Species complex (sample origin documented *n* = 8/10)				
Total other-SC	3	2	2	1
Total *Fusarium* (sample origin documented *n* = 160/182)				
Total *Fusarium* species	11	37	50	62

FO-SC = Fusarium oxysporum-Species Complex; FF-SC = Fusarium fujikuroi-Species Complex; FS-SC = Fusarium solani-Species Complex; SC = Species complex.

**Table 2 jof-07-00246-t002:** Comparison of the performance of three different databases of MALDI-TOF (matrix assisted laser desorption ionization-time of flight) mass spectrometry and NCBI-BLAST (National Center for Biotechnology Information—Basic Local Alignment Search Tool) for the identification of 182 *Fusarium *spp. isolates collected in 20 European centers during a one-year prospective study.

	Gold Standard Identification	Nb of Isolates	Assessment	Ncbi-Blast Approach	MS-Maldi Biotyper Database	MS-MSI-1 (09/2017)—Database	MS-MSI-2 (02/2020)—Database1
*Fusarium dimerum* -SC	*Fusarium dimerum*	4	Correct SP	4	4	4	4
			Correct SC	4	4	4	4
			Discordant/unidentified	0	0	0	0
*Fusarium fujikuroi -SC*	*Fusarium acutatum*	1	Correct SP	0	0	0	1
			Correct SC	1	1	1	1
			Discordant/unidentified	0	0	0	0
	*Fusarium andiyazi*	2	Correct SP	1	0	2	1
			Correct SC	2	0	2	2
			Discordant/unidentified	0	2	0	0
	*Fusarium lactis*	1	Correct SP	1	0	0	1
			Correct SC	1	0	1	1
			Discordant/unidentified	0	1	0	0
	*Fusarium proliferatum*	39	Correct SP	39	24	35	36
			Correct SC	39	24	39	39
			Discordant/unidentified	0	15	0	0
	*Fusarium sacchari*	3	Correct SP	0	0	3	3
			Correct SC	0	2	3	3
			Discordant/unidentified	3	1	0	0
	*Fusarium verticillioides*	8	Correct SP	8	5	8	7
			Correct SC	8	5	8	8
			Discordant/unidentified	0	3	0	0
	Total FFSC	54	Correct SP	49	29	48	49
			Correct SC	51	32	54	54
			Discordant/unidentified	3	22	0	0
*Fusarium incarnatum/equiseti -SC*	*Fusarium equiseti*	2	Correct SP	2	1	2	2
			Correct SC	2	1	2	2
			*Discordant/unidentified*	0	1	0	0
	*Fusarium incarnatum*	2	Correct SP	2	0	1	0
			Correct SC	2	1	1	2
			Discordant/unidentified	0	1	1	0
	Total FIESC	4	Correct SP	4	1	3	2
			Correct SC	4	2	3	4
			Discordant/unidentified	0	2	1	0
*Fusarium oxysporum -SC*	*Fusarium curvatum*	4	Correct SP	0	0	0	0
			Correct SC	4	3	2	4
			Discordant/unidentified	0	1	2	0
	*Fusarium elaeidis*	1	Correct SP	0	0	0	0
			Correct SC	1	1	1	1
			Discordant/unidentified	0	0	0	0
	*Fusarium gossypinum*	1	Correct SP	0	0	0	0
			Correct SC	1	1	1	1
			Discordant/unidentified	0	0	0	0
	*Fusarium languescens*	1	Correct SP	0	0	0	0
			Correct SC	1	1	0	1
			Discordant/unidentified	0	0	1	0
	*Fusarium nirenbergiae*	12	Correct SP	0	0	0	0
			Correct SC	12	5	11	12
			Discordant/unidentified	0	7	1	0
	*Fusarium triseptatum*	1	Correct SP	0	0	0	0
			Correct SC	1	0	1	1
			Discordant/unidentified	0	1	0	0
	*Fusarium veterinarium*	45	Correct SP	0	0	0	0
			Correct SC	45	32	35	44
			Discordant/unidentified	0	13	10	1
	Total FOSC	65	Correct SP	0	0	0	0
			Correct SC	65	43	51	64
			Discordant/unidentified	0	22	14	1
*Fusarium redolens -SC*	*Fusarium redolens*	1	Correct SP	1	0	0	0
			Correct SC	1	0	0	0
			Discordant/unidentified	0	1	1	1
*Fusarium sambuccinum -SC*	*Fusarium culmorum*	1	Correct SP	1	0	0	0
			Correct SC	1	0	0	0
			Discordant/unidentified	0	1	1	1
*Fusarium solani -SC*	*Fusarium falciforme*	8	Correct SP	1	0	3	3
			Correct SC	8	5	8	8
			Discordant/unidentified	0	3	0	0
	*Fusarium keratoplasticum*	8	Correct SP	0	0	6	6
			Correct SC	8	5	8	8
			Discordant/unidentified	0	3	0	0
	*Fusarium lichenicola*	1	Correct SP	0	0	1	1
			Correct SC	1	0	1	1
			Discordant/unidentified	0	1	0	0
	*Fusarium petroliphilum*	14	Correct SP	0	3	12	14
			Correct SC	14	11	14	14
			Discordant/unidentified	0	3	0	0
	*Fusarium solani new-sp1*	8	Correct SP	0	0	0	0
			Correct SC	8	3	8	8
			Discordant/unidentified	0	5	0	0
	*Fusarium solani new-sp2*	2	Correct SP	0	0	0	0
			Correct SC	2	2	2	2
			Discordant/unidentified	0	0	0	0
	*Fusarium solani.sp. robiniae*	2	Correct SP	0	0	0	0
			Correct SC	2	0	2	2
			Discordant/unidentified	0	2	0	0
	*Fusarium solani-FSSC5*	8	Correct SP	0	0	0	0
			Correct SC	8	4	8	8
			Discordant/unidentified	0	4	0	0
	*Fusarium solani-FSSC9*	2	Correct SP	0	0	0	0
			Correct SC	2	0	2	2
			Discordant/unidentified	0	2	0	0
	Total FSSC	53	Correct SP	1	3	22	24
			Correct SC	53	30	53	53
			Discordant/unidentified	0	23	0	0

The gold standard for identification was defined by a DNA phylogenetic tree approach. Numbers of correct identifications at the species level (correct SP), at the species complex level (correct SC), discordant identifications and under threshold identifications are exposed for each of the 27 species. MS: Mass Spectrometry. MBT: MALDI Bio-typer Research Use Only and filamentous fungi combined (Bruker); MSI-1: mass spectrometer identification database (operating between July 2017 and 2020); MSI-2: mass spectrometer identification database (operating since 2020 at https://msi.happy-dev.fr/, accessed on 22 January 2020).

## Data Availability

Data (raw spectra and DNA sequences) can be obtained freely by contacting the corresponding author.

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
