# Peer review of "Clinical Origin and Species Distribution of Fusarium spp. Isolates Identified by Molecular Sequencing and Mass Spectrometry: A European Multicenter Hospital Prospective Study"

_jof, 2021, doi:10.3390/jof7040246_

Round 1

Reviewer 1 Report

Fungal species belonging to the Fusarium genus are responsible for infections in plants, animals and humans. Fusarium spp. are among the dangerous plant pathogens with a high toxicity potential. Secondary metabolites of these fungi, such as deoxynivalenol, zearalenone and fumonisin B1 are among five most important mycotoxins on a European and world scale. Mycotoxins can pose risks to organisms and lead to huge economic losses. These metabolites can cause poisoning the so-called “mycotoxicosis”, posing a considerable threat for humans and animals. In human, Fusarium spp. illnesses encompass a variety of superficial, invasive or disseminated pathologies called fusariosis. The epidemiology of human fusariosis is cloudy due to the rapidly evolving taxonomy.

Authors this manuscript confirmed that the use of mass spectrometry and particularly the MSI online application is a fast and reliable tool for the accurate identification of Fusarium spp. isolates. Moreover, this study allowed assessing the abilities of miscellaneous techniques to identify Fusarium isolates at the species level. The identification was performed by PCR-sequencing and phylogenic-tree approach then used as a gold standard reference for the evaluation of mass spectrometry.

The reviewed manuscript was prepared with great care and its content contains a lot of valuable information. The structure of the paper is correct. All tables and figures are clear, understandable and necessary. The references are sufficient and necessary. 

The overall quality of the manuscript is very good. I think that this manuscript can be published in Journal of Fungi without any changes.

The paper needs some editorial corrections.

Reviewer 2 Report

Can you comment on the fact that Fusarium solani seemed to be underrepresented in the "invasive" group?

Isn't it inevitable that the MSI will appear to be the better database, since it is the main source?

Although this is a multinational study, it seems there is a very strong representation of France. Any insights on Fusarium epidemiology in the different countries or other parts of the world?

Minor comments:

Legend for table 1 is duplicated

I would write the "zeroes' in Table 2-- it will make it more legible and emphasize the point

Reviewer 3 Report

The authors of the manuscript entitled “Clinical origin and species distribution of Fusarium spp. Isolates identified by molecular Sequencing and Mass Spectrometry: a European multicenter hospital prospective study” aimed to study the distribution of Fusarium spp. isolates in a multicenter prospective study. Sequencing methods and Mass Spectrometry were used for strain identification.  Globally, the work is updated and of general interest for the journal readers. However, some issues, especially regarding language must be address in order to clarify and turn the manuscript more scientifically sound.

Line 55 -56  - please replace “… approach then used…” by “…approach. Both methods were used as gold standard for the evaluation of mass spectrometry performance”.

Line 57 – Please replace “satisfying” by “satisfactory”.

Line 73 – Please replace “forms” by “disease”.

Line 96-98 – Please replace by “Due to the relative scarcity of Fusarium infections, relevant studies reporting fusariosis epidemiology data are rare, retrospective, and most of them have a large period to obtain enough… “.

Line 100 – Please delete “originating”.

Line 107 – Please replace “to” by “in”.

Line 109 – “some information” Which information? Please specify.

Line 117-113 – Very confuse! Please rephrase.

Line 113 – Please replace “Clinical conclusion” by “patient clinical outcome”.

Line 114 – Please replace “carriage” by “colonization”.

Line 117 – “contact lens liquid”?! Was this analysed?

Line 176 – Table title is repeated. Please include abbreviation information in the table legend.
